# Testing predictive coding theories of autism spectrum disorder using models of active inference

Tom Arthur[1,2]*, Sam Vine[1], Gavin Buckingham[1], Mark Brosnan[2], Mark Wilson[1], David Harris [ORCID][1]*

1 School of Public Health and Sport Sciences, Medical School, University of Exeter, Exeter, United Kingdom,
2 Centre for Applied Autism Research, Department of Psychology, University of Bath, Bath, United Kingdom

* T.T.Arthur@exeter.ac.uk (TA); D.J.Harris@exeter.ac.uk (DH)

**Data Availability Statement:** All relevant data and code is available online from: https://osf.io/5k48n/. Plotted curves are available at https://osf.io/r9gxf, Autism Spectrum Quotient questionnaire available

## Abstract

Several competing neuro-computational theories of autism have emerged from predictive coding models of the brain. To disentangle their subtly different predictions about the nature of atypicalities in autistic perception, we performed computational modelling of two sensori-motor tasks: the predictive use of manual gripping forces during object lifting and anticipatory eye movements during a naturalistic interception task. In contrast to some accounts, we found no evidence of chronic atypicalities in the use of priors or weighting of sensory information during object lifting. Differences in prior beliefs, rates of belief updating, and the precision weighting of prediction errors were, however, observed for anticipatory eye movements. Most notably, we observed autism-related difficulties in flexibly adapting learning rates in response to environmental change (i.e., volatility). These findings suggest that atypical encoding of precision and context-sensitive adjustments provide a better explanation of autistic perception than generic attenuation of priors or persistently high precision prediction errors. Our results did not, however, support previous suggestions that autistic people perceive their environment to be persistently volatile.

## Author summary

Predictive processing theories of the brain propose that the brain is fundamentally a prediction machine, constantly generating expectations about incoming sensory information. According to these theories, perception and cognition involve the brain's efforts to minimize prediction errors by updating internal models and refining predictions based on incoming sensory data. Several competing accounts of autism have hypothesised that key features of autistic behaviour can be explained through differences in these predictive mechanisms. But these theories propose only subtle differences in predictive mechanisms so are hard to distinguish between. To compare them, we performed computational modelling of two data sets–grip forces during object lifting and eye movements during interception–to understand the differences in prediction between autistic and neurotypical participants. We found no evidence for generic difficulties with prediction but found

at https://osf.io/6szf5; presentation sequences available at https://osf.io/ewnh9/.

**Funding:** This work was supported by the Economic and Social Research Council [grant number: ES/P000630/1], with TA receiving a South-West Doctoral Training Partnership PhD studentship, and a Leverhulme Early Career Fellowship awarded to DH. The funders had no role in study design, data collection and analysis, decision to publish, or preparation of the manuscript.

**Competing interests:** The authors have declared that no competing interests exist.

that individuals with autism face challenges in adaptively adjusting learning rates in response to environmental changes (volatility).

## Introduction

There is substantial evidence of atypical sensory processing in autism (for review see [1]). Disturbances to sensory perception affect the majority of autistic people [2] and have been observed across auditory, visual, and tactile modalities [3]. Additionally, autistic people often have difficulties with performing motor skills. Such movement-related features were initially reported as a general 'clumsiness' in children [4,5], yet an array of long-term issues with motor planning and execution have since been identified [6–11]. Although sensory and motor differences can contribute to significant daily living difficulties in autism, the precise mechanisms that underlie these features remain unclear.

In recent years, a number of promising hypotheses based on the Predictive Processing Framework (PPF; [12–14]) and predictive coding theories of neuronal message passing [15–17] have emerged. These ideas have significant potential for advancing our understanding of clinical neuropsychology (see [18]). Work within the PPF has shown that sensorimotor behaviours are guided by generative models and the drive to minimise prediction error (e.g., [19–21]). Probabilistic inferences about the causes of sensory inputs are made by combining descending (top-down) predictions with ascending (bottom-up) prediction errors that signal deviations from those predictions [17,21–24]. In addition to minimising prediction errors by constantly revising internal models of the world, agents will also act on their surrounding environment to reduce its uncertainty and/or fulfil prior expectations [22,24]. This extension of predictive coding is known as *active inference* and effectively reframes motor control as the fulfilment of counterfactual proprioceptive predictions [25,26].

Several theoretical accounts of autism have adopted this framework to explain sensory atypicalities in autism (for review see [27]). We focus on the following three approaches:

i. Pellicano and Burr's [28] *attenuated-prior account* suggests that perceptual differences in autism result from atypicalities in either the construction of prior expectations about sensory input or in the combining of priors with new sensory information. In essence, priors in autistic people are seen to be overly flat or weak and therefore have an attenuated impact on sensory processing. A similar description is provided in Brock's [29] *bottom-up* account, which claims that the weight afforded to sensory information is atypically high (i.e., precise) in autism. This gives rise to similar predictions as the attenuated prior account, with perception heavily influenced by current state observations, so for simplicity was not included as a separate theory.

ii. Van de Cruys et al [30]. argue that there is a *uniformly high and inflexible level of precision* assigned to prediction errors in autism. These imbalances are said to stem from neurophysiological differences in the extraction of goal-relevant contextual information (i.e., the sensory cues that are used to estimate precision).

iii. Lawson et al [31]. propose that *the encoding of precision is aberrant* in autistic people due to impaired neuro-modulatory gain control. Instead of having persistently attenuated prior beliefs or uniformly high prediction errors, suboptimal control of precision can affect how prediction errors are adjusted in a context-sensitive and iterative manner. Lawson et al [32]. further built upon this idea to propose that autism is characterised by *overestimations of environmental volatility*. At an implicit level, autistic individuals are seen to treat

their surroundings as more unstable than neurotypical individuals, leading to atypical sensory perception.

Although these accounts are all plausible from behavioural and neurological perspectives, it is empirically challenging to isolate many of their claims [27]. For instance, it can be difficult to distinguish attenuated priors from increased sensory precision [29], or atypical prediction error signalling from impaired contingency learning [33]. Moreover, recent attempts to evaluate these models have returned inconsistent results, with many studies reporting that prediction-related functions are *not* impaired in autism (e.g., [34–37]). Indeed, prediction-related differences in autism are typically consigned to more complex or uncertain conditions [33,38]. To date, however, studies testing these theories have tended to use simple associative learning tasks that inadequately capture the complex and dynamic associations that exist within natural environments [27]. Therefore, it is unknown whether these proposed mechanisms are driving autism-related sensory and motor behaviours in more complex movement tasks.

To untangle these competing theories of autism, we adopted computational approaches that are becoming increasingly influential within clinical neuropsychology [39–41]. Constructing generative models to explain the underpinning mechanisms of a system allows us to draw additional inferences from observed behaviours. Instead of merely measuring motor atypicalities in autism, we can estimate the inference processes that may be responsible for those differences. To this end, we describe modelling results from two previously reported data sets that examined sensorimotor atypicalities in autism [42,43]. These two data sets–examining predictive grip forces during object lifting and anticipatory eye movements during manual interception–were chosen as they represent realistic movement skills with important predictive elements that allowed us to probe the mechanisms of active inference (i.e., role of prior beliefs and precision weighting) across visual and motor modalities.

## Experiment 1 –object interaction

The size-weight illusion (SWI) is a paradigm that allows us to observe the influence of predictions on perception and action [44,45]. Due to the feedforward, predictive nature of how we grip and lift objects [46], learned associations between an object's size and weight bias predictive fingertip and lifting forces. As a result, heavy-looking objects are lifted with more force than novel light-looking objects, irrespective of how much they actually weigh [44,45]. SWI studies typically involve collecting both verbally reported perceptions of weight and predictive patterns of fingertip grip and lifting force activity (e.g., [47]). Here we focused solely on the peak rate of change of the application of grip force, which occurs prior to any experience of object weight and therefore provides an uncontaminated index of participants' expectations about object weight that can be used to model active inference. To investigate the mechanisms through which autistic people process and act upon sensory information, we modelled the dynamic updating of motor predictions across each trial of a SWI experiment. In this task, beliefs about the relationship between object size and weight would be updated iteratively over time, enabling the extraction of key parameters relating to active inference theories (e.g., prior beliefs, precision estimates, and multi-level prediction errors).

Previous object lifting studies have returned conflicting results regarding predictive processing theories of autism. Buckingham et al [44]. reported correlations between autistic-like traits and reduced sensorimotor prediction, but these effects did not replicate in Arthur et al. [48] when predictions were related to material properties, rather than object size cues. Furthermore, a follow-up study by Arthur et al [42]. found that autistic individuals did not differ from neurotypical participants in their anticipatory fingertip force profiles, suggesting that there were no generic attenuations in the use of sensorimotor predictions. Further investigation into

the precise mechanisms that underpin these effects is, therefore, required. We modelled the trial-to-trial updating of predictive lifting forces to examine if any differences in belief updating, rates of learning, and perceptions of volatility were present in this dataset that could differentiate between competing theories in the field.

## Materials and methods (Expt 1)

### Ethics statement

The study received approval from the School of Sport and Health Sciences Ethics Committee (University of Exeter, UK) and Department of Psychology Ethics Committee (University of Bath, UK). Written informed consent was obtained in accordance with British Psychological Society guidelines, and the study methods closely adhered to the approved procedures and the Declaration of Helsinki.

Methods and procedures are as described in Arthur et al [42]. and summarised below.

### Participants

Fifty-eight participants, 29 with a clinical diagnosis of autism (19 male, 10 female; 21.28 ± 3.63 years; 25 right-handed), took part in the study. Additional details of diagnostic and inclusion criteria are outlined in Arthur et al [42]. The autism group was compared to a group of neurotypical participants (19 male, 10 female, 21.31 ± 3.30 years; 25 right-handed) that were individually matched based on age, gender, and dominant hand. Three participants in the autism group were excluded from analyses because of invalid grip force data.

As the required sample size for the current analysis could not be determined a priori, a sensitivity analysis was run to estimate the types of effect we were powered to detect. The sensitivity analysis suggested that for the independent group comparisons we had 85% power for effects of $d$ = ~0.8, but only 50% power for smaller effects of $d$ = ~0.5, given the 55 participants included in the analyses (plotted power curves are available from: https://osf.io/5k48n/). The effect sizes that we observed (see *Results and Discussion*) were mostly much smaller than $d$ = ~0.5. It is therefore possible that some differences may have emerged in a much larger sample, but the practical significance of differences of this magnitude is not clear.

### Apparatus and stimuli

Participants lifted four 7.5-cm tall black plastic cylinders using an aluminium and plastic lifting handle fitted with an ATI Nano-17 Force transducer (recording at 500Hz). Objects had two levels of physical diameter (small: 5 cm, large: 10 cm) and two levels of mass (light: 355 g, heavy: 490 g), creating four 'test' items. An additional medium-sized 'control' object (diameter: 7.5 cm; mass: 490 g) was used for practice trials.

### Procedures

Participants were asked to lift and hold the object at a comfortable height above the table surface, using the thumb and forefinger of their dominant hand. The onset (cue to lift) and offset (cue to put down) of each trial was indicated by two computer-generated auditory tones, separated by 4 seconds. Participants were instructed to lift objects in a 'smooth, controlled and confident manner', and to 'gently place the object back on its starting platform'. Firstly, participants completed five 'baseline' trials with the medium object, followed by 32 'test' trials with the four experimental stimuli in one of three pseudorandomized orders (i.e., eight lifts per object). These predetermined trial sequences ensured that each 'heavy' item was lifted at least once before any 'light' trials. Objects were concealed during the resetting of each trial.

After each lift, participants were asked to verbally report a numerical judgement representing how heavy the object felt (larger numbers indicating heavier weights). Following Buckingham et al [44]. no constraints were placed on these values to minimize ratio scaling biases.

## Measures

**Perceived heaviness scores.**   Heaviness ratings were converted to z-scores to place all participants' responses on a common scale (as in [44]).

**Force data.**   Data from the force transducers attached to the top of the objects was used to index the implementation of predictions by the motor system, with stronger grip forces taken to be indicative of an expectation of heaviness, as extensively shown in previous literature [45,47]. Forces perpendicular to the surface of the handle were defined as 'grip force' and were smoothed using a 14-Hz Butterworth filter and then differentiated with a 5-point central difference equation to determine peak grip force rate (pGFR). For the purposes of fitting the learning models, pGFR was then discretised. Grip force rate on each trial was subtracted from the final lift of the baseline trials so that greater forces were taken as an expectation of heaviness and vice versa. We report only the pGFR, not the load force rates, based on the reporting of Arthur et al [42]. that the pGFR outcome was more sensitive to prediction-related differences.

**Computational modelling.**   Over successive lifts, peak grip force rates vary as expectations about p(weight|size) are updated. Fitting learning models enables us to infer the relative balance of priors to new sensory observations during this process. While the principles of hierarchical Bayesian learning are well-founded [23] we first tested whether Bayesian inference provided a good description of grip force updating in our participants, or if it could be more parsimoniously explained by simple associative learning (as in [32,49]). To do this, we compared two families of learning model based on either associative learning (Rescorla-Wagner or Sutton K1) or Bayesian inference (the Hierarchical Gaussian Filter [HGF]; [50]; see Fig 1). We adopted the meta-Bayesian "observing the observer" framework [51] which uses two model components (see Fig 1A)–the *perceptual model*, and the *decision* or *response model*. The perceptual model estimates the agent's perception of their environment (posterior estimates), while the response model estimates the mapping between beliefs and observed actions. Using Bayesian model inversion, the competing learning models were fitted to the trial-to-trial grip force data. We then formally compared the plausibility of the various models using random effects Bayesian model selection (BMS) to identify a generative model which may underlie active inference in our paradigm. Finally, we compared parameter estimates extracted from the winning model between autistic and neurotypical groups.

**Hierarchical Gaussian filter models.**   The HGF model is conceptually related to the "Bayesian brain" hypothesis [23] which proposes that neural and cognitive processing principles should approximate the statistical optimum, i.e., Bayesian inference. In dynamically changing environments, the brain must infer not only the hidden states of the world and how they generate sensory input, but also how those relationships might change over time. This is achieved through hierarchical representations of probability that encode beliefs about the world, the (un)certainty of those beliefs, and how likely the world is to change [54]. The HGF is a generative model in that it attempts to describe the intervening processes that explain how an agent receives a time series of inputs to which it reacts by emitting a time series of responses [50,53]. Crucially, when both inputs and responses are known, the parameters of the perceptual and the response models can be estimated by inverting the model to infer participant-specific parameters and belief trajectories [50]. In the perceptual model, agents make an inference about some parameter $x$ from a series of observations $(u^{(1)}, u^{(2)},\ldots,u^{(n)})$ that provide

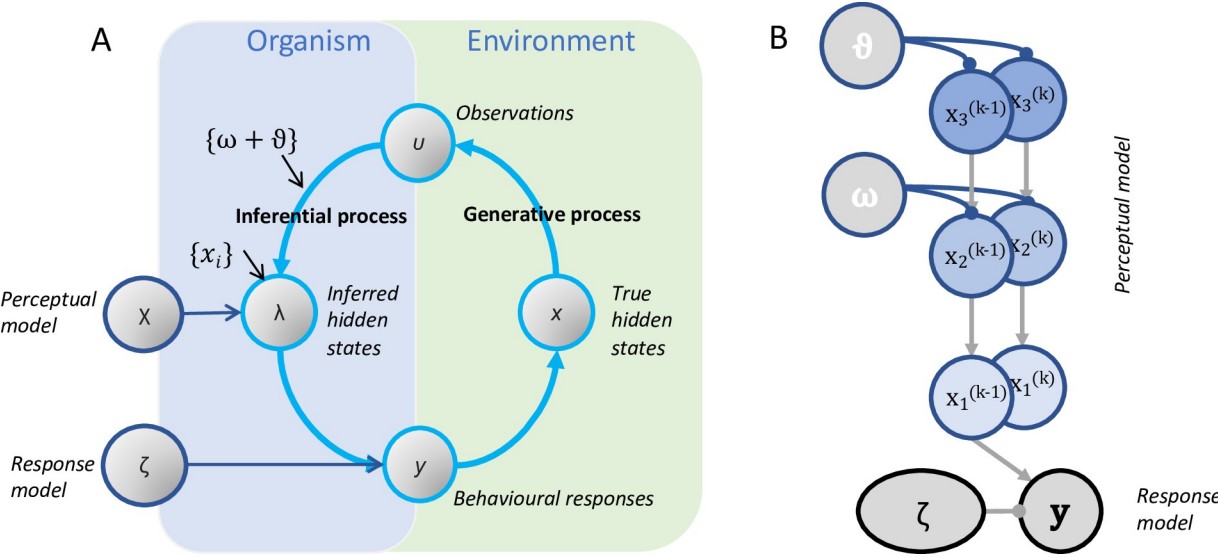

**Fig 1. Basic HGF framework. Panel A** shows the conceptualisation of an agent connected to the world by the sensory information it receives (**u**) and the actions it takes (**y**), as described in active inference. Beliefs about the world depend on inferences about true states (**x**) based on sensory input. The sensory (**u**) and active (**y**) states correspond to the 'Markov blanket' which connects the agent with, but also distinguishes it from, the surrounding environment [52]. **Panel B** shows the basic structure of the HGF. The HGF estimates the inference processes that best explain the behavioural responses of an agent given a time series of inputs [50,53]. Model parameters can be estimated by inverting the model to infer participant-specific parameters and belief trajectories [50]. The perceptual model ($\chi$ in Panel A) is described via beliefs (**x**) represented at multiple layers that evolve across time (**k**), scaled by variance parameters ($\omega$, $\vartheta$). In B, the variance parameter $\vartheta$ controls the rate of change of change in $x_3$, while $\omega$ controls the rate of change in $x_2$.

information about the hidden state. Here, $x$ relates to the unknown weight of the object and $u^{(n)}$ are observed weights. Beliefs about the state $x$ are modelled as a 'Gaussian random walk', which describes the evolution of a time series via a Gaussian probability distribution over $x$. The values of $x$ can be described as follows:

$$x^{(k)} \sim N(x^{(k-1)}, \vartheta), k = 1, 2, \ldots$$

Where $k$ is a time index, $x^{(k-1)}$ is the mean of the distribution at the preceding timepoint and $\vartheta$ is the variance of the random walk. The HGF perceptual model assumes that this variance is is determined by beliefs at the next highest level. For instance, the variance in a participant's belief about the likely weight of an object, after seeing its size, is controlled by a higher level belief about how variable the relationship between size and weight typically is. The coupling between levels is controlled by parameters that shape the influence of uncertainty on learning in a subject-specific fashion. Updates to beliefs about x then proceed according to Bayes theorem, where the prior belief over x is updated with new observations, weighted by their precision. The response model simply contains a constant free parameter ($\zeta$) that represents the 'inverse decision temperature'. This parameter controls the extent to which mapping from beliefs to responses is fully deterministic or highly exploratory ($\zeta$ equates to the shape of the sigmoid mapping from $\mu_2$ to y, i.e., $p(y = 1) = s(\mu_2^{(k-1)}, \zeta)$). Additional details of the mechanics of the model are described in the supplementary files (see: https://osf.io/5k48n/) or in[50].

**Associative learning models.** While there is good evidence that people engage in approximate Bayesian inference, it is important to consider that learning could be better explained by a simpler model. We therefore assessed two simple associative learning models which postulate that agents learn to take actions that maximise the probability of future rewards [55]. These

were the commonly used Rescorla-Wagner (R-W) learning rate model plus the Sutton K1 model [56]. R-W learning models propose that predictions about a value ($v$) are updated over trials ($k$) in proportion to the size of the preceding prediction error ($\delta$) and a stable learning rate scalar ($\alpha$):

$$\Delta v^k \propto \alpha \delta^k$$

While the RW model assumes fixed domain-specific learning rates, the Sutton K1 model assumes variable learning rates that are scaled by recent prediction errors [56]. In these models, the impact of the prediction error is dependent on the magnitude of the error (and previous errors for Sutton K1), rather than flexible precision-weighting based on the strength of priors, likelihoods, or volatility beliefs.

**Model fitting and comparison.** The model building process consisted of first determining starting parameters for the possible models, fitting the data to all possible models and comparing the fits, then extracting the parameters of interest from the winning model. The open source software package TAPAS (available at http://www.translationalneuromodeling.org/tapas; [57]) and the HGF toolbox [50,53] were used for both model fitting and comparison.

For modelling, observations ($u$) were coded to correspond to p(weight|size) (a similar approach to [49]), such that large heavy-feeling and small light-feeling objects indicated that size did predict weight, while large light-feeling or small heavy-feeling objects were deviations from this relationship. Grip forces ($y$) were then median split, such that higher pGFR for larger objects and lower pGFR for small objects indicated an expectation that weight was determined by size.

All models contained free parameters that could vary to accommodate the observed data that we wished to estimate. These parameters were optimised using maximum-a-posteriori estimation to provide the highest likelihood of the data given the model and parameter values. For associative learning models, the free values were beliefs about p(weight|size) and learning rate, which were set at a neutral starting value and given wide variance. For un-bounded parameters in the HGF models we chose prior means that represented values under which an ideal Bayesian agent would experience the least surprise about its sensory inputs, based on a running a simulation with the real sequences from the experiment. The priors were given a wide variance to make them relatively uninformative and allow for substantial individual differences in learning (for additional details on starting priors, and tests of parameter recoverability and identifiability, see https://osf.io/rbtqp).

After each of the possible models had been fit to the observed participant data, they were compared via Bayesian model selection [58], using spm_BMS.m routines from the SPM12 toolbox (https://www.fil.ion.ucl.ac.uk/spm/software/spm12/). Bayesian model selection treats the model as a random variable that could differ between participants, with a fixed (unknown) distribution in the population. It provides an estimate of the probability that a given model outperforms all others in the comparison (the 'protected exceedance probability').

## Results and discussion (Expt 1)

### Model comparison

We examined four types of learning model: two versions of the HGF (3-level [HGF3] and 4-level [HGF4]) were compared to two associative learning models (Rescorla-Wagner and Sutton K1). We also computed a second subtype of all four models, using 'felt' inputs instead of 'veridical' inputs. Grip and lifting forces are believed to be distinct from the perceptual illusion of felt heaviness in the SWI paradigm [45,47,59]. We therefore assumed that veridical inputs (i.e., the true weight of the objects) would provide the best model of predictive grip forces. However, as subjective heaviness ratings (i.e., 'felt' inputs) could potentially provide an

explanation of fingertip forces, we also created models with heaviness ratings as the input ($u$). While these two versions could not be compared directly, if log-model evidences for veridical inputs were consistently higher than for felt inputs, then this would provide some confirmation that veridical inputs allowed a better model of changes in fingertip forces.

As predicted, results of the model fitting and comparison showed that the veridical input variants had higher log-model evidence than the felt model in all cases (Fig 2A–2C). Parameter comparison (below) was therefore conducted on the veridical type models. Results strongly indicated that the HGF4 was the most likely model (for both veridical and felt types), with a protected exceedance probability of 1.00 in both cases (see also plot of model probabilities in Fig 2C). This winning model contained an additional level of hierarchical beliefs compared to the more common HGF3, which could reflect the highly uncertain task environment. In this task, participants did not know what kinds of objects to expect or in what order they might be received. Indeed, participants are often unsure how many different objects they have been lifting in SWI experiments, so their beliefs about volatility may well have been unstable, hence the need for the additional level in the model.

Notably, plots of log model evidence (see Fig 2A) indicated that both the 3- and 4-level Bayesian models provided a better explanation of the data than the two associative learning models. This indicates that participant's trial-by-trial object lifting behaviours were successfully explained via the general principles of active inference and hierarchical predictive processing. Parameters of the winning HGF4 model could next be compared between groups, to examine how underlying sensorimotor control mechanisms may differ in autistic individuals.

## Parameter comparison

Model parameter comparisons generally did not show any autism-related differences in the dataset (see Fig 3D). Independent t-tests showed no significant differences in beliefs about p (weight|size) [$t(53) = 1.37$, $p = .18$, $d = 0.37$] and no differences in the decision temperature parameter zeta [$t(53) = 0.07$, $p = .95$, $d = 0.02$]. This lack of differences in the use of prior expectations supports the main findings reported in Arthur et al [42]. and adds to growing evidence that sensorimotor difficulties cannot be accounted for by Pellicano and Burr's [28] 'hypopriors' theory (e.g., see [37,60–62]).

Secondly, the present results suggest that there were no atypicalities relating to autistic learning rates (i.e., the way in which prior expectations about p(weight|size) were updated over time). There were no differences in the parameters governing the random walk at level two (i.e., $\omega_2$) [$t(53) = 0.55$, $p = .59$, $d = 0.15$] or level three (i.e., $\omega_3$) [Welch's $t(39.4) = 0.33$, $p = .74$, $d = 0.09$]. There was no variance in $\omega_4$ so no comparison was run. There was also no difference in learning rates ($\alpha$) at the second [Welch's $t(42.7) = 1.29$, $p = .21$, $d = 0.35$], third [$t(53) = 0.57$, $p = .57$, $d = 0.15$], or fourth [$t(53) = 0.25$, $p = .80$, $d = 0.07$] model levels. This suggests that the way in which autistic individuals weighted new sensory evidence (relative to priors) was consistent with patterns shown by neurotypical individuals, contrary to the high and inflexible precision of prediction errors hypothesis [30].

In contrast to Lawson et al. [32], we found no evidence that autistic individuals perceived the volatility of their environment differently from their neurotypical counterparts. Indeed, independent t-tests showed no significant between-group differences in beliefs about volatility ($\mu_3$) of p(weight|size) [$t(53) = 0.55$, $p = .59$, $d = 0.15$], or volatility of the volatility ($\mu_4$) [Welch's $t(40.3) = 0.29$, $p = .77$, $d = 0.08$]. We cannot, however, draw firm conclusions about volatility beliefs at this stage, given that there were minimal changes in environmental contingencies within this experiment. As such, we now turn to the issue of perceived volatility and context sensitivity in experiment 2, to directly examine Lawson's conceptual model.

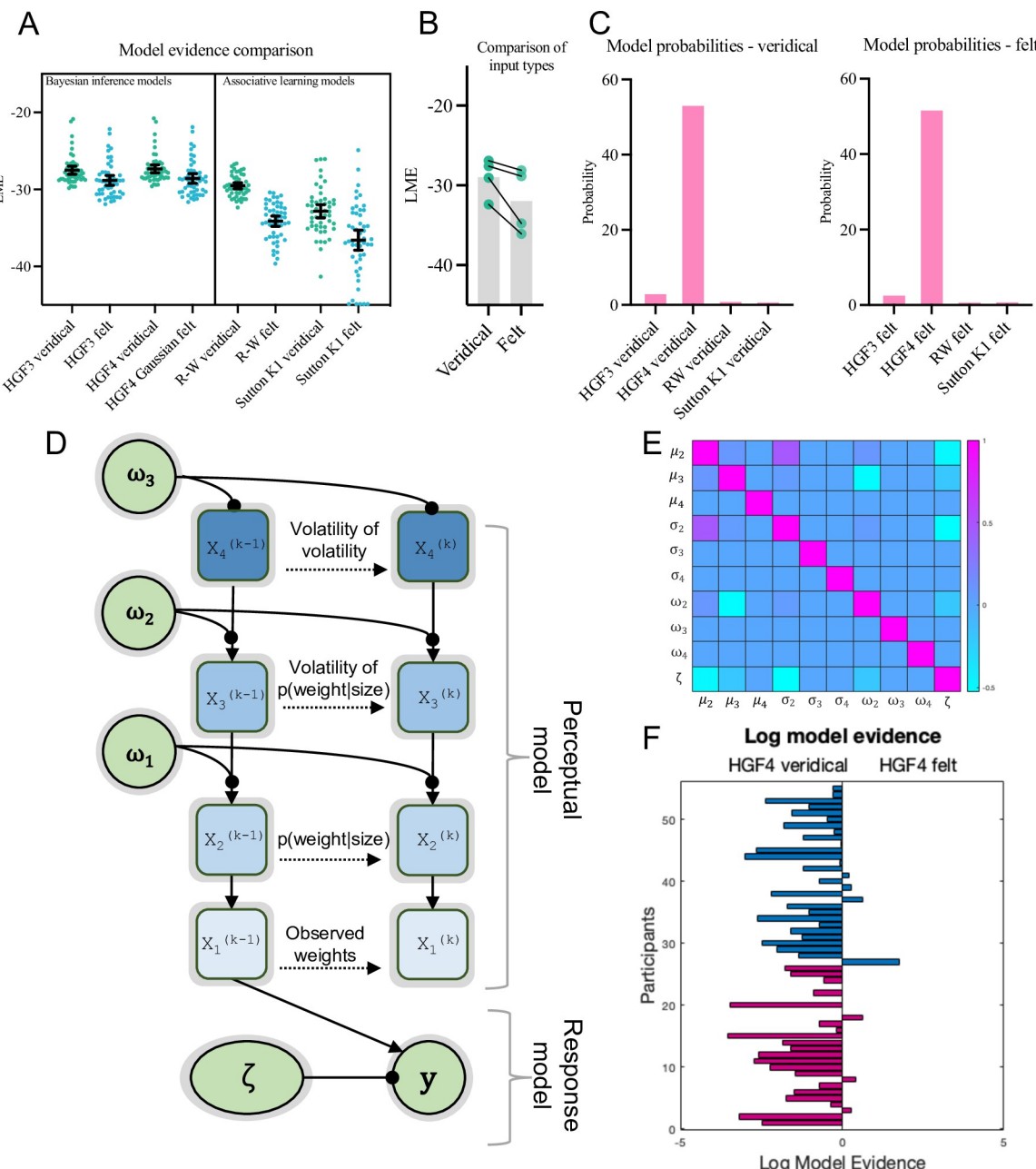

**Fig 2. Results of the model fitting. Panels A-C** show the results of the model fitting and comparison procedures, illustrating the log-model evidence (LME) for all models (**A**) and for the veridical compared to felt models (**B**). **Panel C** illustrates the probabilities of the different models in the participant population based on Bayesian model selection, where HG4 was the most likely generative structure. **Panel D** shows a schematic of the winning HGF4 model. **Panel E** shows the parameter identifiability matrix for the HGF4, which shows that no model parameters were highly correlated (i.e., one could not simply be substituted for another). **Panel F** shows a comparison of the log model evidence per participant between the veridical and felt versions of the HGF4, illustrating the veridical version was better for most participants.

## Experiment 2 –interceptive actions

Experiment 2 applied a similar modelling approach–this time using eye movement data during interceptive actions–to further examine autistic sensorimotor behaviours. Arthur et al. [43] report data from a virtual reality (VR) tennis task in which participants were required to

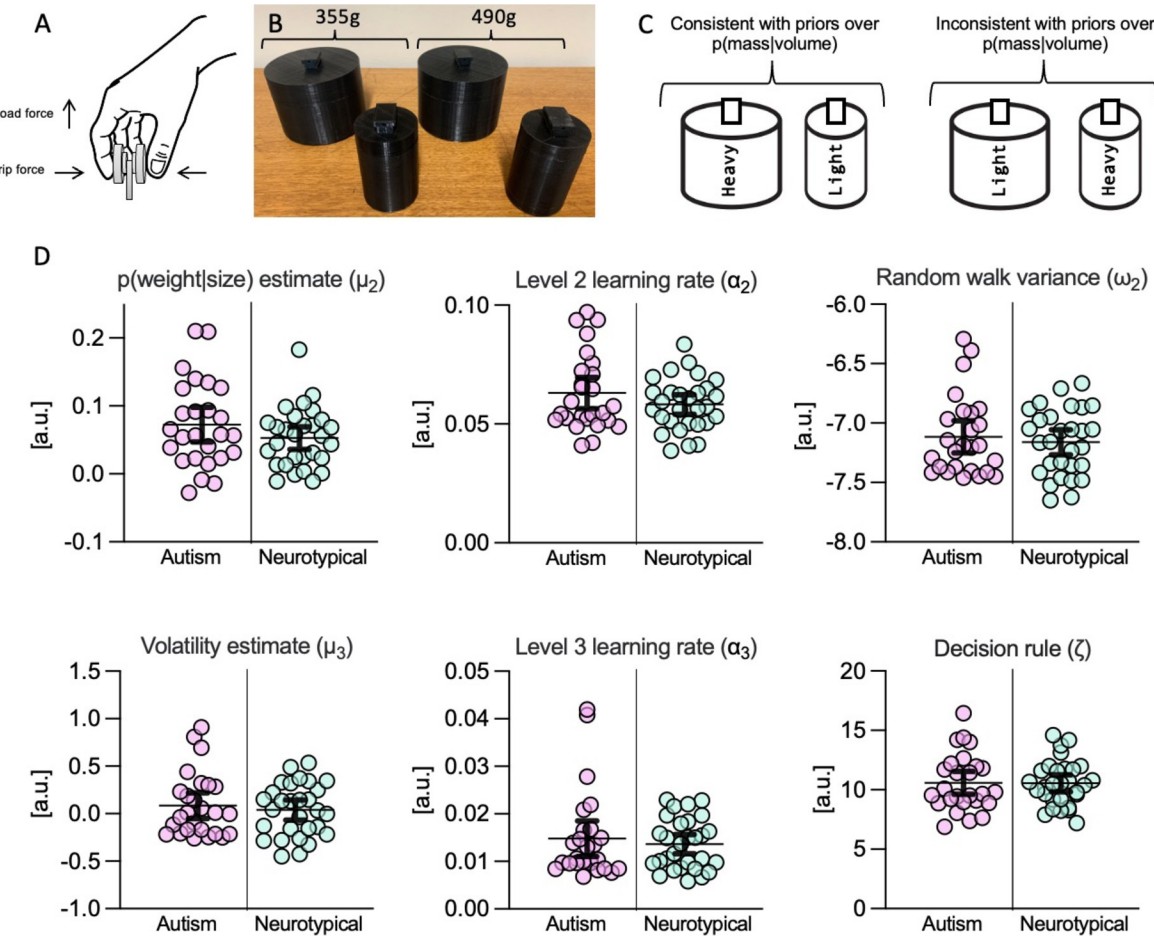

**Fig 3. Results of the parameter comparison. Panel A:** Illustration of grip and load force on the force transducers. **Panel B:** The stimulus objects. **Panel C:** Illustration of the relationship between mass and volume in the stimuli. **Panel D:** Plots show individual data points with mean and 95% confidence intervals for comparisons of model parameters between autistic and neurotypical groups.

intercept a ball that bounced in front of them. In this task, the relative weighting of prior beliefs and prediction errors can be uniquely assessed by controlling the presentation order of balls with more or less predictable post-bounce trajectories. In contrast to Expt 1, the authors manipulated these features *over time*, creating experimental conditions that fundamentally differed in terms of stability and volatility. As such, this data set allowed us to model active inference and volatility processing in a more complex movement skill that is representative of the tasks that autistic individuals might find difficult in the real-world.

In active inference, vision is used to reduce expected prediction errors by actively sampling the visual scene to minimise surprise [21], with fixations and saccades cast as experiments driven by our hypotheses about the world [63]. During the interception of a bouncing ball, a visual fixation is generally directed towards the bounce point of the oncoming projectile [64]. The exact spatial location of this fixation is known to be sensitive to prior beliefs about ball bounciness, as it is shifted to a higher location when higher bounces are expected [65]. The position of this fixation is also sensitive to wider probabilistic context (volatility) of the environment [19,43]. Much like predictive gripping forces in Expt 1, the way in which the fixation location is adjusted and updated over sequential trials is, therefore, indicative of beliefs about outcomes and can be used to model active inference. Notably, Arthur et al. reported that both

autistic and neurotypical participants used prediction-driven strategies in this task, by anticipating the future bounce location of the ball. Neurotypical individuals were found to adjust their gaze location between stable and volatile conditions, but autistic participants did not display these context-sensitive behaviours and effectively treated the stable and volatile conditions as similarly uncertain.

In their original analysis, Arthur et al [43]. interpreted the above findings as evidence that autistic individuals overestimate the volatility of their environment, as suggested by Lawson et al [32]. The precise mechanisms that underpinned these effects did, however, ultimately remain unclear, as no direct measures of volatility beliefs were taken. We therefore used computational modelling approaches to better understand the generative processes underlying these findings and directly test competing predictive coding theories of autism. In relation to the theories discussed previously, Pellicano and Burr's [28] attenuated-prior account suggests that autistic eye movements should be less driven by previous experiences for all trials in this task. Van de Cruys et al.'s [30] hypotheses would additionally imply that, due to increased precision of prediction errors, autistic individuals should consistently update their beliefs (i.e., the spatial location of predictive bounce fixations) more readily than neurotypical individuals. Although increased learning rates could also be explained by Lawson et al.'s [31] theoretical account, this hypothesis does not predict generic differences in prior beliefs or prediction error weighting, as differences in predictive processing are context-dependent. Instead, atypical *adjustments* in sensory sampling behaviours would be expected under conditions of varying uncertainty or volatility. Moreover, on the basis of Lawson et al. [32], we would expect autistic individuals to increasingly attribute unexpected events to changes in their surrounding environment (i.e., they should show increased learning rates at higher levels of the HGF).

## Materials and methods (Expt 2)

### Participants

The sample (described in [43]) was made up of 90 participants, 30 with a clinical diagnosis of autism. A large neurotypical sample was recruited in the original study to provide sufficient power for correlational analysis (e.g., as in the *Supplementary Materials*), hence the inequality in the group sizes. This ratio of autistic to neurotypical participants aligns with clinical research recommendations (i.e., between 1:2 and 1:4) where a larger control group can help improve statistical power (see [66]). For the present analyses, a series of further criteria were applied to ensure we were using accurate and reliable gaze recordings for the modelling. Individual datasets were re-inspected and participants who were missing >15% of gaze fixation values (due to missing trials, loss of tracking, or a lack of predictive eye movements) were excluded from analyses. A sample of 59 participant datasets were subsequently deemed appropriate for the modelling (36 male, 23 female; 21.73 ± 4.54 years; 51 right-handed); 17 had a formal diagnosis of autism, and the remaining 42 were age-matched neurotypical controls.

As the required sample size could not be determined a priori for the current modelling work, we conducted a sensitivity analysis to estimate the types of effect we were powered to detect. The sensitivity analysis suggested that to detect an interaction effect in a 2 (group: autism v neurotypical) x 2 (condition: stable v volatile) ANOVA we had 85% power for conventionally large effects of $f = \sim0.4$, but only 50% power for more moderate effects of $f = \sim0.25$, given the 59 participants (plotted power curves are available in the supplementary files: https://osf.io/r9gxf). Many of the effect sizes that we observed (see *Results and Discussion*) were in this medium to large range, so we were adequately powered for many of these tests.

Participants were naïve to the aims of the experiment and reported no prior experience of playing VR-based racquet sports. The study received approval from the School of Sport and

Health Sciences Ethics Committee (University of Exeter, UK) and Department of Psychology Ethics Committee (University of Bath, UK). Informed consent was obtained in accordance with British Psychological Society guidelines, and the study methods closely adhered to the Declaration of Helsinki.

## Apparatus and stimuli

A full description of the experimental set-up and materials can be found in [43]. To summarise, we developed a VR simulation of a racquetball court, using the gaming engine Unity (Unity Technologies, San Francisco, CA). This environment was presented on an HTC Vive VR system (HTC Inc., Taoyuan City, Taiwan), which recorded movements of a headset and hand controller at 90 Hz. The VR headset included a Tobii eye-tracking system, which employs binocular dark pupil tracking to monitor gaze at 120 Hz (spatial accuracy 0.5–1.1˚; latency 10 ms). To establish tracking accuracy, gaze was calibrated over five virtual locations. These calibration procedures were performed before both study conditions and upon any obvious displacement of the headset during the experiment.

During each trial, a ball was launched from a height of 2m at the front of the virtual racquetball court (see Fig 4A), which was just above an aiming target of five concentric circles. Participants stood in the centre of the court, approximately 9m from the front wall. They then attempted to hit the projected balls, which looked like real tennis balls (5.7 cm in diameter), back towards the target using a virtual racquet that was animated in the virtual world by tracking the hand-held VR controller. The virtual racquet was $0.6 \times 0.3 \times 0.01$ m, but the collision area associated with it was exaggerated by 20 cm to enhance the detection of ball-to-racquet collisions.

**Procedures.**   After arriving at the laboratory, participants provided informed consent and completed the Autism Spectrum Quotient questionnaire (AQ-26; described in supplementary materials: https://osf.io/6szf5). Next, they were introduced to the VR and completed six familiarisation trials. Individuals were instructed to hit virtual balls back towards the centre of the projected target, and that the appearance of the balls would be cued by 3 auditory tones. They were told that the ball should bounce once, but that they were free to intercept it at any point after this event. All shots were forehand swings and ball bounces were accompanied by auditory feedback. No visual, proprioceptive, or verbal feedback was available on racquet-ball contact. Instead, a neutral 'pop' sound was incorporated to minimise the impact of feedback on performance.

Crucially, the projected balls had two distinct physical profiles which permitted precise control over participants' experience of *expected* and *unexpected* events. All balls followed the same pre-bounce trajectory and speed along the midline of the room (vertical speed: -9 m/s at time of bounce), but ball elasticity was either consistent with the 'tennis ball-like' appearance (elasticity set at 65%) or was unexpectedly high (set at 85%). Unexpected trials consisted of an abrupt change in 'bounciness' that would deviate from any real-world prior expectations about natural ball materials. The expected and unexpected trial events were ordered into two distinct blocks, to generate either *stable* (in which balls were presented in 'predictable' serial orders), or *volatile* (where the ground truth probability regularly switched) conditions. Both contained the same number of expected ($n = 30$) and unexpected ($n = 15$) trials. However, under stable conditions, the marginal likelihood of facing a 'normal' ball remained fixed at 67.67%. Conversely, these likelihoods were unstable in the volatile condition, and switched irregularly between highly- (83%), moderately- (67%) and non-predictive (50%), in blocks of 6, 9 or 12 trials (presentation sequences are available at https://osf.io/ewnh9/). Each condition was separated by a short break, with the order of the stable and volatile blocks counterbalanced across participants.

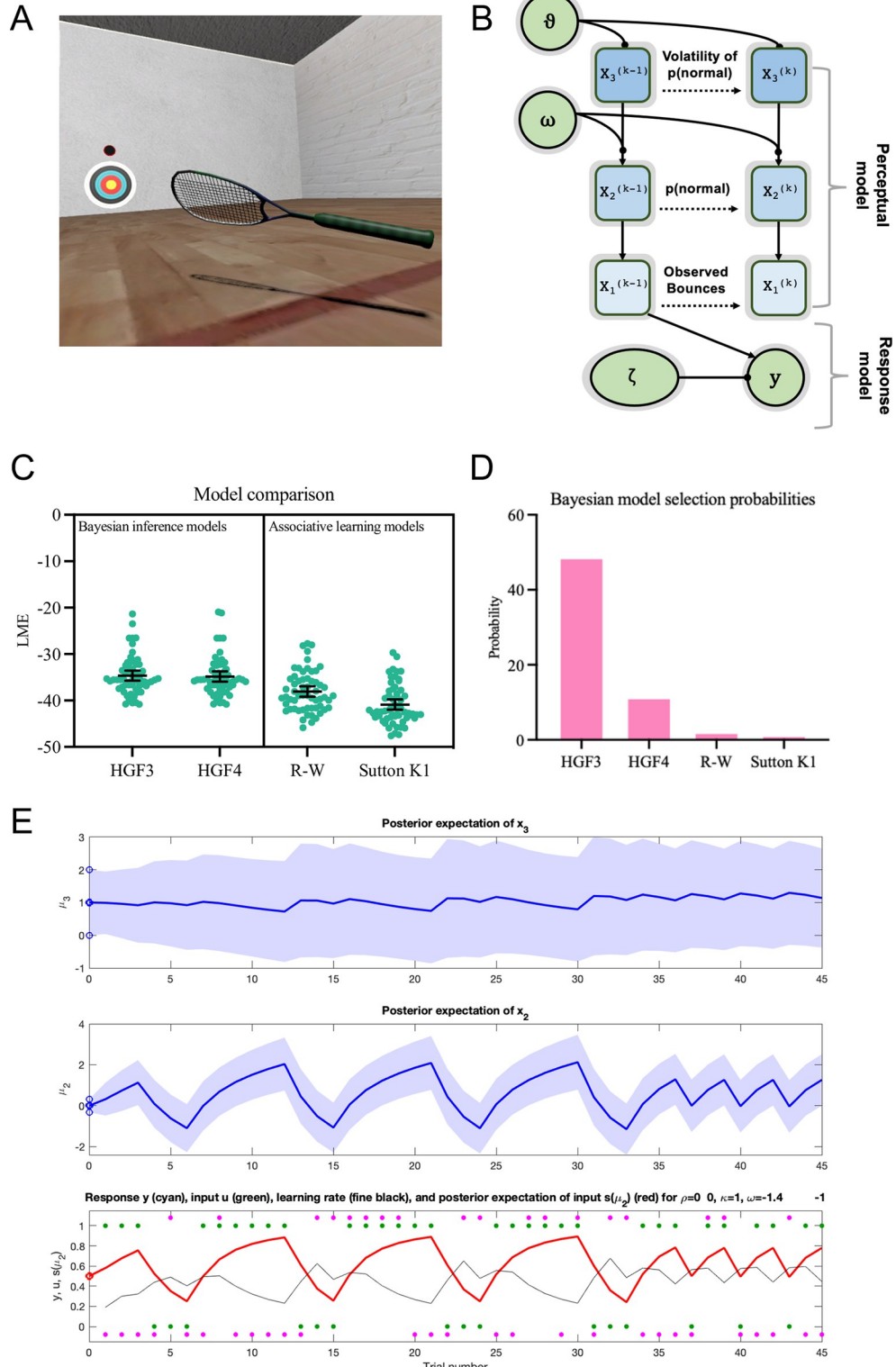

**Fig 4. Results of the model fitting. Panel A** shows the task environment and **Panel B** shows the winning HGF3 model structure. **Panels C and D** illustrate the log-model evidence (LME) for all models and the probabilities of the different models in the participant population as indicated by Bayesian model selection. **Panel E** shows the belief trajectory of the fitted model for a single participant. In the bottom section, observed outcomes are shown in green (1 = normal ball, 0 = bouncy). The inferred posterior belief about the likelihood of a normal or bouncy ball (i.e., s(μ₂))

is in red and the binary distributed response variable (pitch angle: above or below median) in pink. The thin black line shows the learning rate. Mean and 95% CIs for $\mu_2$ and $\mu_3$ are shown in blue in the middle and top panels.

**Measures.**  *Gaze Pitch Angle*. A range of measures were reported in the original study [43]. Here, we only report the 'gaze pitch angle' as an indicator of prior beliefs, for the purposes of modelling predictive sensorimotor behaviours (as in [19]). As discussed, individuals generally direct a single fixation to a location a few degrees above the bounce position of an oncoming ball [67,68]. The vertical position of this fixation (the *gaze pitch angle*) is sensitive to beliefs about likely ball trajectories, with fixations directed to a higher location when larger bounces are expected. As the fixation occurs before the bounce is observed, the fixation location is driven by an agent's predictions about ball elasticity and thus provides an indicator of their prior beliefs over time.

A single unit gaze direction vector was extracted from the inbuilt eye-tracking system in the VR headset, with features defined according to head-centred, egocentric coordinates. The extracted gaze vector and the ball-relative-to-head position were then plotted in 2D space, to create relative 'in-world' angular metrics. All trials were segmented from the moment of ball release until the time frame corresponding to ball contact. Trials with > 20% missing data, or where eye-tracking was temporarily lost (> 100 ms) were excluded. Gaze coordinates were treated with a three-frame median filter and a second-order 15 Hz Butterworth filter [69,70]. A spatial dispersion algorithm was then used to extract gaze fixations [71]. These were operationalised as portions of data where the point of gaze clustered within 3˚ of visual angle for a minimum duration of 100ms [72]. We extracted the fixation position at the moment of bounce (expressed as gaze-head pitch angle). Data values that were > 3.29 SD away from the mean were classed as outliers ($p < .001$), and participants with > 15% of data identified as missing and/or outliers were excluded.

**Computational modelling.**  We adopted the same framework to model the inference processes associated with the updating of the bounce fixation over successive trials as we did for grip forces in experiment 1. To determine whether Bayesian inference or simple associative learning was the more likely generative process, we again compared the fits of two HGF models (HGF3 and HGF4) against two associative learning models (R-W and Sutton K1). To fit learning models to the behavioural data from the interception task, the pitch angle variable (eye position just before the bounce point) was converted into a discrete variable: when gaze was shifted to a higher location than on the previous trial (>1SD change) this was taken as a shift towards higher p(expected) and vice versa. This approach was chosen to mirror i) the approach taken in Expt 1 and ii) that taken in a previous paper using this task [19]. Eye position can alternatively be modelled as a continuous variable, and simply requires a different linking function between beliefs and actions in the HGF [50]. Recent work has also begun to model complex motor actions (e.g., arm movements) using a continuous active inference approach (see [40]).

## Results and discussion (Expt 2)

### Model comparison

We evaluated the fits of the four models to the behavioural data across all participants using Bayesian model selection [58]. As in experiment 1, LMEs were higher for both the HGF3 and HGF4 compared to the two associative learning models (see Fig 4C). This showed that the models adopting flexible hierarchical weighting of prediction errors better explained participants' gaze data than simple associative learning computations. The HGF3 (see Fig 4B) was

deemed to be the most likely model, with a protected exceedance probability of 1.00 (Fig 4D). This model was therefore chosen as the most likely generative process and the participant-wise parameter estimates were extracted to examine how active inference processes may differ between autistic and neurotypical individuals. The fact that the HGF3 outperformed a HGF4 model solution in this experiment suggests that beliefs about volatility were relatively stable within the two study blocks. When compared to the illusory object lifting conditions in Expt 1, we speculate that participants were more certain about the variety of ball bounce outcomes and trial order changes in the virtual racquetball task.

## Parameter comparison

As is evident in Fig 5, there were differences in variance between the Autism and Neurotypical groups for some parameters. As ANOVA is typically described as robust to deviations from normality [73], we retained a parametric approach for the omnibus test for most variables. Non-parametric tests were used for $\alpha_3$ which was visibly strongly skewed. For follow-up comparisons we used non-parametric Welch's t-tests where there were between group differences in variances.

The first parameter of interest was $\zeta$, the decision rule parameter, which indicated the degree to which responses were determined by beliefs or more noisy/exploratory policies. A 2 (group) x 2(condition) mixed model ANOVA showed no significant group [$F(1,57) = 0.01$, $p = .91$, $\eta_p^2 = 0.00$], condition [$F(1,57) = 0.39$, $p = .54$, $\eta_p^2 = 0.01$], or interaction [$F(1,57) = 1.52$, $p = .22$, $\eta_p^2 = 0.03$] effects, suggesting no differences in participants' response models.

Next, we examined the posterior state estimates for $x_2$ and $x_3$ ($\mu_2$ and $\mu_3$). ANOVA indicated no significant condition [$F(1,57) = 0.45$, $p = .45$, $\eta_p^2 = 0.01$] or interaction [$F(1,57) = 0.01$, $p = .91$, $\eta_p^2 = 0.00$] effects for $\mu_2$, the posterior expectation of ball outcome. There were, however, group differences in $\mu_2$ [$F(1,57) = 6.60$, $p = .01$, $\eta_p^2 = 0.10$], with higher values in the neurotypical group (Fig 5C) indicating that autistic individuals held weaker expectations about receiving a 'normal' ball than their neurotypical counterparts (in line with [28]).

There was no group level difference in $\mu_3$ [$F(1,57) = 2.28$, $p = .14$, $\eta_p^2 = 0.04$], the posterior state estimate of environmental volatility. There was, however, an overall effect of condition [$F(1,57) = 5.37$, $p = .02$, $\eta_p^2 = 0.09$] and the interaction effect was close to the significance threshold [$F(1,57) = 3.84$, $p = .055$, $\eta_p^2 = 0.06$]. Follow-up Welch's t-tests with Holm-Bonferroni correction indicated that there was no significant difference between groups in the stable condition [$t(56.7) = 0.77$, $p = .44$, $d = 0.19$], but large differences during the volatile trials [$t(57.0) = 3.00$, $p = .008$ $d = 0.72$] (see Fig 5D). In these trials, neurotypical participants appeared to perceive greater volatility than their autistic counterparts, following context-sensitive adjustments in their posterior state estimates. As illustrated in Fig 5D, the autism group did not display these contextual adjustments, indicating that they were less inclined to update their state beliefs between stable and volatile conditions.

Next, we examined rates of belief updating (learning) at levels $x_2$ and $x_3$, referred to as $\alpha_2$ and $\alpha_3$ (see Fig 5E and 5F). For $\alpha_2$, there were clear group level differences [$F(1,57) = 6.40$, $p = .01$, $\eta_p^2 = 0.10$], but condition [$F(1,57) = 0.73$, $p = .40$, $\eta_p^2 = 0.01$] and interaction effects [$F(1,57) = 1.46$, $p = .23$, $\eta_p^2 = 0.03$] were not statistically significant. Although the interaction was not statistically significant, there appeared to be a divergence pattern in the plot (Fig 5E) so we performed post-hoc tests to check if the group differences were similar across conditions. Welch's t-tests with a Holm-Bonferroni correction, confirmed that there was no significant difference between groups in the stable condition [$t(49.7) = 1.31$, $p = .20$, $d = 0.34$], but a significant difference was present in volatile trials [$t(52.8) = 3.09$, $p = .006$, $d = 0.73$]. Here, a much higher learning rate was observed for neurotypical participants, indicating that volatility-

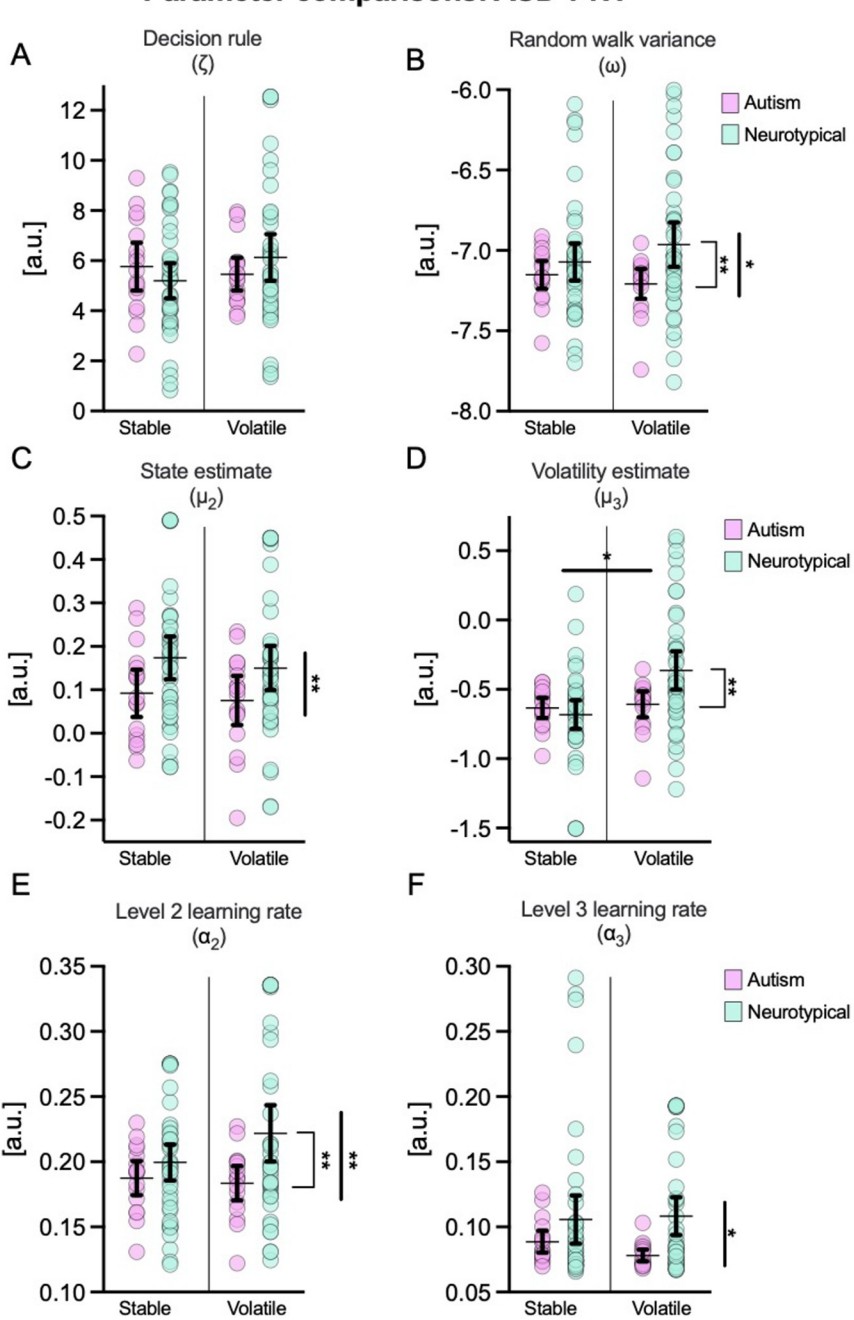

**Fig 5. Mean and 95% CIs of model-derived estimates.** Figures show the decision parameter, $\zeta$, (**panel A**), the random walk variance parameter, $\omega$, (**panel B**), posterior beliefs about $x_2$ (ball bounciness; **panel C**) and $x_3$ (environmental volatility; **panel D**), and learning rates at levels 2 (**panel E**) and 3 (**panel F**) of the HGF. Thicker significance bars indicate group-level effects and thinner bars indicate pair-wise differences. *$p < .05$, **$p < .01$, ***$p < .001$.

related increases in learning rate were not displayed by autistic individuals. For $\alpha_3$, a Friedman's test showed no effect of condition [$\chi^2 = 1.37$, $p = .24$], but a Kruskal-Wallis test indicated an overall effect of group [$H(1) = 3.90$, $p = .048$]. Follow-up Mann-Whitney tests indicated that there was no difference between groups for the stable condition [$z = 376$, $p = .76$, $r_{rb} =$

.05], but that learning rate was higher for neurotypical participants in the volatile condition [$z = 238$, $p = .047$, $r_{rb} = 0.33$].

Next, we examined ω (Fig 5B) and ϑ, the variance parameters in the random walk at $x_2$ and $x_3$. For ω, effectively an estimate of environmental volatility, ANOVA showed an overall effect of group [$F(1,57) = 4.75$, $p = .03$, $\eta_p^2 = 0.08$], but no effect of condition [$F(1,57) = 0.13$, $p = .72$, $\eta_p^2 = 0.002$], and no interaction [$F(1,57) = 1.30$, $p = .26$, $\eta_p^2 = 0.02$]. Welch's t-test showed no significant differences between autism and neurotypical groups in the stable condition [$t(56.2) = 1.13$, $p = .26$, $d = 0.28$], but a large difference in the volatile trials [$t(57.0) = 3.00$, $p = .008$, $d = 0.72$]. Specifically, the high levels of environmental instability on these trials were detected less readily by autistic compared to neurotypical individuals. For ϑ, the variance in the dispersion of the volatility state estimate ($x_3$) (i.e., the volatility of volatility), ANOVA showed null group [$F(1,57) = 1.84$, $p = .18$, $\eta_p^2 = 0.03$], condition [$F(1,57) = 0.01$, $p = 0.91$, $\eta_p^2 = 0.00$], and interaction [$F(1,57) = 0.03$, $p = .86$, $\eta_p^2 = 0.00$] effects.

As autism is generally viewed as a spectrum disorder, we also conducted a supplementary analysis (available in the online files: https://osf.io/6szf5) to examine the relationship between model parameters and autistic traits as measured by Autism Spectrum Quotient questionnaire. Correlations were weak suggesting the observed effects were related to diagnostic status more than behavioural traits.

## General discussion

Recent neuro-computational theories have identified a variety of mechanisms that could explain atypical sensory processing [1] and movement control [8] in autism, but these theoretical accounts are yet to be adequately tested using the types of naturalistic behavioural tasks that autistic people find challenging (but see [74] for comparison of Bayesian theories of autism in a purely perceptual task). To probe the generative processes responsible, we applied a generative modelling approach to active inference behaviours from both an object lifting paradigm and an interceptive movement task. Our results have important implications for each of the main theoretical explanations of autism, which we outline below.

The attenuated-prior account [28,29] characterises perceptual and behavioural differences in autism as resulting from overly flat or weak priors, which exert a chronically diminished influence on sensory processing. Contrary to these claims, our models of predictive grip force rates (Expt 1) indicated no difference in beliefs about the relationship between the size and weight of objects. Findings from other perceptual and motor tasks have also questioned the 'hypopriors' hypothesis (e.g., [37,42,48,60–62]). We did, however, observe weaker expectations about 'normal' ball bounce profiles in autistic individuals during the more dynamic interceptive task (Expt 2) and weaker priors have been reported in some settings (e.g., [75,76]). These context-dependent results support the generally inconsistent study findings in the field (see [27,33] for reviews) and align with our previous observations [42,43,48,77]. Indeed, although the location of predictive gaze fixations may generally be atypical in autistic individuals during interceptive motor tasks, their visual sampling behaviours are still strongly driven by prior expectations [43] and are sensitive to explicit cues about likely ball bounciness [77]. Hence, there is growing evidence that diminished use of priors is not a satisfactory explanation for the totality of autistic perceptual and motor behaviours. While the integration of prior beliefs and sensory observations is undoubtedly atypical in some tasks or environments, the contextual factors and mechanisms that moderate these hierarchical predictive processes must be accounted for in future theoretical work.

The high and inflexible precision of prediction errors account [30] proposes that bottom-up information from sensory input (and the deviation of sensations from expectations) is

afforded a high and fixed level of precision in autistic people. Consequently, perception is dominated by incoming sensations and internal predictive models are updated at a persistently faster rate. We did not, however, find evidence to support this hypothesis. Firstly, the trial-by-trial updating of predictive grip forces in Expt 1 did not appear to be accelerated or atypical in autism. Secondly, during the interceptive task (Expt 2), there was also no evidence of accelerated updating of anticipatory eye movements in autistic participants. Instead, estimated learning rates at both levels 2 and 3 of the learning model were *lower* than neurotypical equivalents, which suggests that autistic individuals were actually resistant to updating their beliefs in response to precision-weighted prediction errors. Such findings are consistent with recent observations that learning rate is *not* persistently elevated in autistic people [62,78], and indicates that the high precision of prediction errors account may be limited in explaining sensorimotor atypicalities.

Finally, Lawson and colleagues [31,32], have proposed that autism-related differences in sensory processing stems from atypical precision encoding in the brain. Generic strength of priors or weighting of prediction errors are not proposed to be different, but rather it is the context-sensitive mechanisms that regulate how these belief signals are dynamically *adjusted* over time. Notably, Expt 2 revealed clear differences between autistic and neurotypical individuals in this regard. Firstly, while neurotypical participants perceived our volatile study conditions to be more changeable than their stable trial equivalents, autistic individuals' beliefs about environmental volatility were not adjusted between experimental blocks (Fig 5D). Furthermore, under volatile conditions, functional adaptations to learning rate at level 2 (about p (normal)) and level 3 (about volatility) were not evident for autistic individuals, despite being displayed by their neurotypical counterparts (Fig 5E and 5F). These results align with previous observations that sensorimotor processing is insensitive to varying levels of environmental uncertainty in autism [32,42,79]. However, while these results support the idea that context-sensitive modulation of prediction errors and learning rate is atypical in autistic individuals, the effects we observed were in the *opposite* direction to that predicted by Lawson et al [31]. who claim that autistic individuals tend to process sensory environments as persistently volatile. In Expt 1, we found no evidence of differences in volatility estimations between autistic and neurotypical individuals, although the analyses were limited by the restricted variation in volatility in this task. In Expt 2, our model parameters ($\mu_3$ and $\omega$) indicated that posterior estimates of volatility were actually lower in the autism group, compared to neurotypical controls (see Fig 5D). One reconciling explanation, is that if autistic adults perceive the world as *persistently* changeable and unpredictable, they may be less inclined to update their beliefs in response to volatility (i.e., beliefs about volatility could be weighted with atypically high precision). Consequently, Lawson's claim that autistic individuals process the world as persistently volatile could still hold. Given that similar types of 'hyperpriors' have already been proposed as a critical element of mood disorders [80], further research into sensorimotor skills could examine the persistence of participants' volatility beliefs (e.g., volatility of volatility) and/or the mechanisms that underpin precision encoding functions (e.g., using computational modelling or neuro-imaging methodologies).

Nevertheless, some key theoretical issues relating to predictive processing accounts of autism require further examination, and additional questions remain about the applicability of these frameworks across wider daily living behaviours and social domains (for further discussions, see [27,81]). The accounts examined in this paper are primarily rooted in traditional Bayesian brain and predictive processing theories rather than active inference per se. In this, and previous [43,48], work we have attempted to apply these theoretical ideas to more complex motor actions, but there is still work to be done to formalise these theories in terms of free energy minimization and epistemic reward. One option to more directly probe atypicalities in

prediction error signalling while maintaining more naturalistic movement tasks is the use of EEG (e.g., see [82,83]). Such future enquiries are not just significant from a conceptual perspective; they could also implicate future practical support for the autism community. Indeed, by better understanding how sensory information is processed during naturalistic tasks, and by identifying precise neuro-computational differences that underpin autistic-like behavioural traits, one can augment the development of evidence-based support tools and learning provisions (see [84]).

## Conclusions

This paper sheds new light on the underlying mechanisms responsible for sensorimotor atypicalities in autism. Here, the use of computational modelling techniques has added a useful perspective to existing research findings in the field, allowing us to disambiguate between competing theoretical hypotheses. We provide evidence that some key features of autistic sensorimotor behaviour can be effectively explained via the general principles of active inference. Such results contribute to the scientific and aetiological understanding of autism and could represent an important step towards developing appropriate practical support. Specifically, our analysis suggests that autism research must account for differences in the context-sensitive modulation of prediction error in the brain (e.g., [31,32]). This account may be compatible with attenuated priors which are observed in some tasks/contexts. Our results also raise further questions about how autistic people process sensory information in dynamic and unconstrained movement tasks.

## Author Contributions

**Conceptualization:** Tom Arthur, Sam Vine, Gavin Buckingham, Mark Brosnan, Mark Wilson.

**Data curation:** Tom Arthur, David Harris.

**Formal analysis:** Tom Arthur, David Harris.

**Methodology:** Tom Arthur.

**Supervision:** Sam Vine, Gavin Buckingham, Mark Brosnan, Mark Wilson.

**Visualization:** David Harris.

**Writing – original draft:** Tom Arthur, David Harris.

**Writing – review & editing:** Sam Vine, Gavin Buckingham, Mark Brosnan, Mark Wilson.

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
