## [Decision Letter · Decision Letter 0]

13 Jun 2023

Dear Dr Harris,

Thank you very much for submitting your manuscript "Testing predictive coding theories of autism spectrum disorder using models of active inference" for consideration at PLOS Computational Biology.

Your manuscript was reviewed by members of the editorial board and by two independent reviewers. In light of the reviews (below this email), we would like to invite the resubmission of a significantly-revised version that takes into account the reviewers' comments.

We cannot make any decision about publication until we have seen the revised manuscript and your response to the reviewers' comments. Your revised manuscript is also likely to be sent to reviewers for further evaluation.

Sincerely,

Jean Daunizeau

Academic Editor

PLOS Computational Biology

Daniele Marinazzo

Section Editor

PLOS Computational Biology

Reviewer's Responses to Questions

**Comments to the Authors:**

Reviewer #1: This manuscript uses computational modelling (HGF and two associative learning models) to determine whether any of the three leading predictive processing accounts of autism are plausible. The study is well designed and easy to follow, with succinct explanations of the theories being compared. This topic is very important, and the study does a good job at distinguishing the theories computationally, and explaining the results. I am especially appreciative of the use of very specific and relevant behavioural data (including active components, which is essential to the theories of active inference, but often neglected) and would be interested to hear what the authors think about perhaps extending this analysis to similar neural signals in the future. Overall, the manuscript was thorough and clear.

I would like the authors to add some information about whether it is fair to compare models with different inputs (as in the felt and veridical data for the SWI experiment), and what it means for conscious perception vs. action that the preferred model is (maybe?) different between these two measures.

I am concerned about the difference in sample size between groups for the second experiment, and do not see that this is addressed by the analysis. This may impact on the results, as Figure 5 suggests there is far greater variability in the neurotypical group? This should be addressed in the next version of the manuscript.

I would appreciate some discussion about why the two experiments might have different winning models (HGF3 v 4), and why the 4-level model would be preferred to model the data with less temporal complexity. I do like the structure of the general discussion as it stands, so perhaps this could be included in the discussion for Expt 2.

I am unsure why the posthoc analysis was done for an insignificant interaction (Exp2 α2).

I would appreciate a little more discussion and flagging that the volatility group differences for Expt 2 are the opposite of what Lawson et al. (2017)’s theory would predict. This would also be good to note in the abstract, as the results do not seem as clear as it suggests.

I note a few small errors in the manuscript – pg. 15 references Figure 3E – I presume D is meant, as there is no E. Pg.19 says “Autistic Quotient questionnaire” this should read “Autism Spectrum Quotient”. Pg. 20 – I cannot see where the number of blocks is reported, please include this information.

Reviewer #2: General Evaluation: This paper offers an engaging exploration of the predictive processing framework and its utility in understanding sensory and motor discrepancies in individuals with autism. By employing computational approaches to devise generative models, the authors successfully elucidate the system's underlying mechanisms, facilitating an enhanced interpretation of observed behaviours. The structure of the paper is logically organized, and the application of a robust methodology delivers noteworthy insights into the domain of clinical neuropsychology.

Strengths:

- The paper is logically structured, and the research question is unequivocally stated. The authors' detailed explanation of the predictive processing framework and its relevance to understanding autism is praiseworthy.

- The innovative application of computational models to uncover the system's underlying mechanisms adds a fresh perspective to the study, promoting a more profound understanding of the observed behaviours.

- The authors have applied two distinct datasets to investigate sensorimotor atypicalities in autism, bolstering the validity of their findings.

- The provision of data and code on the OSF platform enhances transparency and reproducibility.

Areas for Improvement:

- General:

The relationship between the results and Active Inference rather than Predictive Processing in a conventional sense is somewhat unclear. For instance, the models and conclusions seem to omit Free Energy minimization or explicit epistemic reward (e.g., lines 33-35; 191-194; 379-382; 388-391; 717).

The authors model actions discretely. An explanation and the discussion about how to adopt a continuous approach (as opposed to a median split, lines 264-266) could enrich the paper (lines 724-725).

- Method:

While the methodology section is thorough, it could benefit from a more explicit delineation of the various components in the models. This enhancement could make the study more comprehensible to readers less familiar with this modeling approach.

Figure 5F depicts distributions that are non-Gaussian, suggesting the authors may need to employ non-parametric tests to compare Autism and Neurotypical subjects.

The authors evaluate the statistical power required in both studies but do not specify whether the observed effect sizes align with these estimations.

- Discussion:

The discussion could delve more deeply into the implications of the findings for our understanding of autism and the predictive processing framework. Questions about the relationship with key diagnostic symptoms, particularly social aspects, and the clinical and general applicability of these findings, remain.

The perspectives section should consider referencing neuroimaging and how it could resolve remaining questions (lines 703-705).

- Figures:

The use of bar plots (Fig. 2A & 4C) is not advised as they can be misleading for non-Gaussian distributions. Box plots (like in Fig. 3 & 5) or violin plots might be more appropriate.

The term "cyan" is used in Figure 4E, but "magenta" appears more accurate (both in the Figure itself, and its caption line 551).

- Language:

Please take care with the terminology used when discussing the distinctive aspects of autism, specifically avoiding words like "aberrant" (abstract & line 74).

Recommendation: Minor revisions. This paper is of high quality and contributes valuable insights into the predictive processing framework and its role in understanding autism. Still, the authors should consider addressing the points raised above to enhance the clarity and overall impact of their work.

**Have the authors made all data and (if applicable) computational code underlying the findings in their manuscript fully available?**

Reviewer #1: Yes

Reviewer #2: Yes

PLOS authors have the option to publish the peer review history of their article (what does this mean?). If published, this will include your full peer review and any attached files.

Reviewer #1: **Yes: **Kelsey Perrykkad

Reviewer #2: No
---

## [Decision Letter · Decision Letter 1]

28 Aug 2023

Dear Dr Harris,

We are pleased to inform you that your manuscript 'Testing predictive coding theories of autism spectrum disorder using models of active inference' has been provisionally accepted for publication in PLOS Computational Biology.

Best regards,

Jean Daunizeau

Academic Editor

PLOS Computational Biology

Daniele Marinazzo

Section Editor

PLOS Computational Biology

Reviewer's Responses to Questions

**Comments to the Authors:**

Reviewer #1: My comments have been adequately addressed and the paper has been improved. Congratulations to the authors, I hope to see it published soon!

Reviewer #2: I thank the authors for their diligent efforts in addressing all of my comments to the fullest extent possible.

**Have the authors made all data and (if applicable) computational code underlying the findings in their manuscript fully available?**

Reviewer #1: Yes

Reviewer #2: Yes

PLOS authors have the option to publish the peer review history of their article (what does this mean?). If published, this will include your full peer review and any attached files.

Reviewer #1: **Yes: **Kelsey Perrykkad

Reviewer #2: **Yes: **Guillaume Dumas

---

## [Editor Report · Acceptance letter]

7 Sep 2023

PCOMPBIOL-D-23-00326R1 

Testing predictive coding theories of autism spectrum disorder using models of active inference

Dear Dr Harris,

I am pleased to inform you that your manuscript has been formally accepted for publication in PLOS Computational Biology. Your manuscript is now with our production department and you will be notified of the publication date in due course.

With kind regards,

Judit Kozma
